# A Review of Welfare Assessment Methods in Reptiles, and Preliminary Application of the Welfare Quality^®^ Protocol to the Pygmy Blue-Tongue Skink, *Tiliqua adelaidensis*, Using Animal-Based Measures

**DOI:** 10.3390/ani9010027

**Published:** 2019-01-17

**Authors:** Amelia L. Benn, David J. McLelland, Alexandra L. Whittaker

**Affiliations:** 1School of Animal & Veterinary Sciences, University of Adelaide, Adelaide, SA 5005, Australia; a1686368@student.adelaide.edu.au (A.L.B.); dmclelland@zoossa.com.au (D.J.M.); 2Zoos South Australia, Frome Road, Adelaide, SA 5000, Australia

**Keywords:** reptiles, welfare, play, pygmy skink, assessment tool

## Abstract

**Simple Summary:**

Reptiles are commonly housed in wildlife parks and zoos, and are increasingly being kept as pets. Husbandry of reptiles is complex; signs of pain or disease can be challenging to recognize, and behavior is not always well understood. Therefore, assessment of reptile welfare may be difficult. In comparison to mammals, methods of welfare assessment in reptiles are under-investigated. In this paper we review the literature on welfare assessment techniques in reptiles. We determine that indicators of negative affective state are better characterized in reptiles, and are likely easier to apply in a welfare assessment tool. Indicators of positive affective state such as play behavior, and judgment bias experimental techniques, should be investigated further in reptiles. We also explore the application of the Welfare Quality^®^ Protocol to the pygmy blue-tongue skink, an endangered species. This application presents examples of predominantly animal-based indicators that may be further investigated for use in a tool in this, and other reptile species.

**Abstract:**

Reptiles are held at wildlife parks and zoos for display and conservation breeding programs and are increasingly being kept as pets. Reliable indicators of welfare for reptiles need to be identified. Current guidelines for the captive management of reptiles utilize resource-based, rather than animal-based indicators; the latter being a more direct reflection of affective state. In this paper we review the literature on welfare assessment methods in reptiles with a focus on animal-based measures. We conclude that, whilst a number of physiological and behavioral indicators of welfare have been applied in reptiles, there is need for further validation of these methods across the diversity of species within the Class. Methods of positive welfare state assessment are comparatively understudied and need elucidation. Finally, we examine some widely-used welfare assessment tools in mammals and explore the application of the Welfare Quality^®^ Protocol to the endangered pygmy blue-tongue skink, *Tiliqua adelaidensis*. We propose that this framework can form the basis for the development of taxon-specific tools with consideration of species-specific biology.

## 1. Introduction

There is increasing interest in, and focus on, the welfare of zoo animals, both within and outside the industry. Zoos are increasingly positioning themselves, and acting as conservation and education institutions. These activities are necessarily underpinned by zoo collections, which frequently include reptiles. Ensuring that the welfare of animals in zoos is optimized is increasingly a focus of the industry itself, and of the general public. As such, monitoring welfare needs to become an institutional priority [1].

Reptiles make up approximately 1.4% of the pet population in the UK [2], and 2.4% in the US [3]. According to Warwick et al. (2017), 75% of pet reptiles in the UK die within their first year at home [4]. This mortality rate suggests knowledge of general husbandry and health care is commonly lacking, in addition to these species’ inherent low adaptabilities to artificial environments [5]; the importance of optimizing a reptile’s affective state would also likely be underappreciated.

Access to adequate nutrition, environmental conditions and the opportunity for behavioral expression are key drivers of an animal’s mental state [6]. As human caretakers, with complete control over captive animals’ care and environment, we have an ethical obligation to safeguard, and indeed promote the welfare of those animals under our care. Reliable tools to objectively assess affective state, and consequently welfare, of reptiles would better allow those caring for reptiles to identify deficiencies in wellbeing and to evaluate management strategies employed to improve health and welfare.

In this paper, we review animal-based measures of affective state in captive reptiles with a focus on practical assessment methods, such as spontaneous behaviors, rather than experimental applications. In addition, we highlight the need for assessment tools to be valid, feasible and reliable. Finally, we explore the application of the Welfare Quality^®^ Protocol to the pygmy blue-tongue skink (*Tiliqua adelaidensis*), an endangered species of reptile. This terrestrial lizard, endemic to Australia [7], was thought to be extinct until it was rediscovered in 1992 [8]. A captive breeding program has been established at Zoos South Australia to support ongoing conservation efforts for the wild population.

## 2. Reptile Diversity

Animals of the Class Reptilia are primarily ectothermic vertebrates, characterized by a covering of scales or scutes, encompassing over 10,000 distinct species [9]. The Class is one of the most diverse classes of living animals, comprising four Orders: Squamata, Rhynchocephalia, Testudines and Crocodilia. Squamata refers to the scaled reptiles consisting of over 8000 species of snake and lizard. These are currently the most common species kept as pets [2]. Rhynchocephalia comprises two nocturnal species of tuatara, which only inhabit the islands of New Zealand and live for 60–70 years. Testudines include all species of turtles, tortoises and terrapins. This Order of marine, freshwater and terrestrial species is characterized by long-life spans and modified rib cages that form an encasing shell. Crocodilia, the most ancient Order of extant reptiles, refers to the 25 species of semi-aquatic crocodiles, alligators, caimans and gharials. The Crocodilia are among the few species of reptile that overtly display advanced maternal care towards their offspring [9].

Reptiles are often seen as ‘behaviorally simple’, making welfare assessment using behavioral expression difficult. This may be due to their ectothermic physiology and low metabolic rate, with prioritization of energy being used for essential behaviors like hunting or basking [10]. Another major limitation for using behavior as an indicator is the lack of field-based observation of reptile behavior. In consequence, the diversity and complexity of reptile social behavior has probably been underestimated, with research tending to focus on other taxa, principally mammals [11]. Accordingly, social systems are currently known in less than 1% of lizard species. This includes behaviors such as parental care, monogamy and kin recognition [12]. Physiological measures of welfare also appear understudied in comparison to other vertebrates, with a seeming lack of baseline reference data for various analytes, and significant variation in values apparent both between and within species. 

The heterogeneity of reptiles makes it challenging to identify reliable indicators of welfare across species, with substantial variation in anatomy, diet, feeding strategy, ecological niche, social structure, reproductive strategy, and so on. Furthermore, the paucity of literature on reptile welfare renders it problematic to draw firm conclusions on this subject. Studies performed on mammalian welfare vastly outnumber those in non-mammalian animals; for example, between 1985 and 2004, studies in mammals represented 92.2% of publications on enrichment and only 0.57% of these studies were on reptiles [13]. However, a database search does suggest that research efforts into reptile welfare have increased in recent years, and it is hoped that this trend continues.

## 3. Trends in Animal Welfare Science

The concept of welfare is defined in terms of affective states, and their balance over time [14]. Affective states are variably described as feelings, emotions or moods, which might include, amongst others, happiness, fear, depression and pain [14]. Affective states comprise behavioral, physiological and cognitive components that have two dimensions, being arousal and valence. Arousal represents the level of bodily activation or strength, for example ‘excited’ compared to ‘relaxed’. Valence refers to the direction of the stimulus such that the stimulus may provoke a positive or negative state [14]. Despite an almost universal acceptance that animals are sentient [15], there remains controversy over the degree to which animals actually ‘feel’ emotions, using cognition to evaluate and reflect on their experiences [16]. Measurement of affective state can be used to infer welfare state. Animals that mostly experience positive affective states, such as happiness, are described as having good welfare, with the converse occurring when animals mainly experience negative states such as fear [17]. Therefore, animal welfare is generally considered as a long-lasting state.

There have been several trends in animal welfare assessment methods. These include a transition from resource to animal-based indicators, and from group to individual assessment. Resource-based measures are recorded in the animals’ environment. They are quantitative, highly repeatable across different observers, and easy to record. However, these measures may not be correlated with the actual affective state or condition of an animal. For example, a lizard may have optimum lighting and space in its enclosure but may be suffering from a debilitating disease. Conversely, animal-based indicators are measured directly in animals, using a combination of physiological, behavioral and health variables, and may consider the individuals’ lifetime experience [18]. This requires observation of the animal and their interaction with the environment, and is generally regarded as a more accurate indicator of welfare. Nevertheless, these measures are time-intensive and difficult to record. Furthermore, they require familiarity with the species of interest, and can lack in repeatability between observers due to their often subjective nature [19]. Consequently, current welfare assessment strategies for reptiles tend to use resource-based measures. In this paper we will focus on review of animal-based measures of affective state. Animal-based measures are also predominantly featured in the proposed welfare indicators for the pygmy blue-tongue skink. 

Perhaps the most significant shift in this field of science is the promotion of positive welfare states over simply minimizing negative states. Mellor & Beausoleil (2015) utilized the Five Domains model, initially created to assess welfare compromise, to consider positive experiences that may enhance welfare. In contrast to aspects of negative welfare, methods for assessing positive affective state are under-researched, especially in reptiles [6]. This is of particular concern when current animal welfare thinking regards consideration of positive affective states as an entrenched element [20]. 

## 4. Assessment of Positive Affective State

Research into negative aspects of animal welfare far surpass that of positive welfare states, and yet one without the other does not provide a holistic understanding of an individual’s welfare. It may be beneficial then to view overall welfare assessment as a continuum rather than simply a ‘good’ or ‘poor’ binary. Current evidence suggests that the evaluation of inputs valued by an animal via positive outcomes is the best method to assess positive welfare [21]. With respect to reptiles, the literature on positive welfare state is mostly focused on assessment of enrichment provision. 

### 4.1. Enrichment

Mellor & Beausoleil (2015) state that positive welfare states are promoted by two methods. Firstly, when survival-related negative affects are minimized [6], that is, essential needs must be addressed before employing enrichment strategies [22]. Secondly, replacing situation-related negative affects with positive affects, namely by providing a stimulus-rich environment [6]. 

Enrichment preference tests are useful to perform with reptiles under human care. However, these are time-intensive and difficult to apply to the wide variety of species held in zoos, let alone the multitude of extant reptiles [23]. Bashaw et al. (2016) used preference testing to examine the response of leopard geckos (*Eublepharis mascularius*) to five types of enrichment. The geckos responded predominantly to thermal and feeding enrichment, perhaps because these address the behavioral needs of this species [24]. This agrees with the idea that effective enrichment strategies should be based on a motivation, rather than being designed to ‘distract’ or ‘entertain’ the animal, as a random novelty might achieve [25]. This result also supports the idea that reptiles respond to enrichment like carnivorous mammals: that they benefit from increased behavioral opportunities. Case et al. (2005) found that eastern box turtles (*Terrapene carolina carolina*) in barren enclosures spent more of their time trying to escape and less time resting, compared to those in enriched housing [26]. However, there have been studies that have not demonstrated benefit from enrichment in reptiles, such as Rosier & Langkilde’s (2011) study on the arboreal eastern fence lizard (*Sceloporus undulates*) [27]. In this study, treatment groups were separated into enclosures that did or did not provide elevated basking platforms. Results showed that there was no effect on survival, baseline plasma corticosterone, activity, basking behavior, time spent hiding or body condition score. The authors suggest that this does not imply that enrichment will not improve welfare for reptiles, but that the effectiveness of species-specific enrichment strategies need to be evaluated. The varying results of providing enrichment to reptiles across different taxa highlight the challenges of generalizing a welfare assessment tool across these different species. Consideration of the indicators in such a tool must be customized to meet species-specific needs.

It has been suggested that live prey would be enriching for carnivorous reptiles, to allow them to hunt and capture their prey [28,29]. However, this poses legal and ethical issues for the prey species, and prey species can potentially injure the reptile or serve as intermediate hosts for parasites [22]. An alternative could be to create movement of dead prey, for example using a mechanical ball or floating a food bowl in water.

It has been suggested that providing an enclosure that mimics a species’ natural environment is enrichment in itself. In a stimulus preference study, the eastern indigo snake (*Drymarchon couperi*), demonstrated a clear preference for river rocks (typical of the species’ natural environment) over polyvinyl chloride pipe and a sandbox, and no interest in a novelty item (gel stickers) [30]. In practice it is virtually impossible to comprehensively replicate the animal’s natural habitat in captivity [22]. Further, in some instances, removing aspects of a species’ natural habitat in captivity will improve welfare, for example, removing of predators by housing individuals in a secure solitary enclosure, and providing veterinary care to minimize the impact of injury and disease. Through continually improving our knowledge of reptile species and their natural habitats, and reducing threats that present themselves in the wild, we can improve enclosure design and husbandry with optimization of animal welfare in mind.

### 4.2. Spontaneous Behavioral Measures 

It is essential to consider the species-specific behavior expressed by an individual to use behavioral indicators of welfare effectively. Behavior alone is not directly indicative of affective state but provides insight into how an animal feels [25]. Exploratory behavior, affiliative behavior, vocalizations, facial expressions, anticipatory behavior and play have all been used for assessment and monitoring of positive emotion and have been investigated in reptiles to varying extents [14,18].

Exploration is considered a behavior with an underlying motivation and can be influenced by positive or negative fear affective states [14]. Many squamate lizards have demonstrated curiosity and highly developed exploratory behavior towards objects. One example is Anole lizards investigating orange or yellow objects with no other apparent purpose [12]. This may not be applicable to other taxa or even between individuals within a taxa, and could instead elicit a stress response. It has been suggested that moving a heat source or opening different retreats enhances exploratory behavior [31]. Hernandez-Divers (2001) states that even a partial alteration of an enclosure may increase exploratory behavior in reptiles, and act as a means for behavioral enrichment [32]. A study by Rose et al. (2014) followed the behavior of two species of reptile, the common chuckwalla (*Sauromalus ater*) and the corn snake (*Pantherophis guttatus*) after performing small changes to enclosure design. These changes elicited increased activity levels in both species of reptile as they utilized the enclosure furnishings provided to them more often then they spent hiding. It was concluded that spatially complex enclosures will increase the behavioral repertoire of reptiles much like other species [33]. Exploratory behavior is perhaps the easiest behavior to observe in a captive reptile. 

Social interactions may also be a good indicator of wellbeing [34]. Affiliative behavior between individuals of a social species can be readily observed, making this a useful indicator for amateur as well as experienced welfare assessors [18]. Researchers often opt to partition vertebrate social behavior into ‘social’ or ‘non-social’ categories, rather than along a spectrum. Reptiles have therefore been placed in the ‘non-social’ group, presumably due to their lack of expression and overt affection, and lack of maternal care in most species. However, current evidence supports a range of behaviors from ‘solitary’ to ‘highly social’ both within and between species [11]. Given the paucity of research into social structure of reptiles, labels such as ‘social’ or ‘non-social’ would be better used to describe individual’s behavior in a particular context, rather than being applied to taxa [35].

Vocalization is also a commonly used indicator of affective state, which can be measured non-invasively. It is often assessed in concert with facial expression [18,36]. It was not until recently that vocal plasticity has been investigated in reptiles, and hence there is limited literature to ascertain its utility in welfare assessment [37]. Facial expression has also been used to assess psychological and emotional experience. This however, has only been indicated in mammals which have homogenous facial musculature [38], for example, the horse [36]. Reptiles have less advanced facial musculature, making it challenging to utilize facial expression as an assessment tool [39]. 

Anticipatory behavior is used to demonstrate positive state associated with reward. It has been correlated with dopaminergic activity, and may be influenced by need or the environment [14]. The dopamine system acts upon previously ‘liked’ rewards, stimulating goal-directed behavior when anticipating a reward that will improve positive affective state [18]. This behavior can be stimulated with positive reinforcement training in reptiles where the consummatory reward may be food or shelter, and may in itself be considered a form of enrichment [31,40]. Anticipatory behavior *per se*, as a measure of positive affect, does not appear to have been studied in reptiles. 

Play has been shown to correlate with increased opioidergic activity [41], but can be associated with negative states, for example, to relieve social tension [18]. ‘Play’ is a concept rarely associated with ectothermic species, as it is metabolically and, potentially unnecessarily, costly [12]. Burghardt et al. (1990) state that reptiles are capable of play and benefit from problem-solving challenges. They give an example of using sensory enrichment using the scent of prey items in objects placed into their enclosure [42]. A study performed in 1996 documents the provision of various objects to a Nile soft-shelled turtle (*Trionyx triunguis*) housed in a zoo. This turtle had a history of self-mutilation that dramatically decreased after providing it with balls, sticks and hoses which it interacted with 20.7% of the time it was observed. Its activity levels also increased to beyond that expected (67.7%). This study suggests that object play is beneficial to captive reptiles that do not have the metabolic demand of those in the wild [43].

Within the Reptilia, play behavior is especially prevalent in the larger, more metabolically active monitors. These reptiles have been observed to push around novel objects and have even played ‘tug of war’ with their keepers with no aggressive undertones [12]. There have been many documented play behaviors in turtles (see [12] for summary). It has been suggested that their increased life spans provide more time to accrue play behavior, and to experience its long-term benefits [12]. These findings suggest play behavior may be a useful welfare indicator, at least in the larger monitor species and turtles. However, given reptile species diversity, play is probably not going to represent a useful animal-based measure to incorporate into an assessment tool for those smaller, less active species.

### 4.3. Judgement Bias Testing & Response to Novelty

Judgement bias studies have been foreshadowed as the most reliable measures of positive affect for species, such as reptiles, where play is not particularly useful [44]. However, these studies fall into the realm of experimental types of welfare assessment tool. Judgement bias testing is based on the premise that an animal experiencing positive affective states will respond to an ambiguous cue with a more optimistic judgement, following training in a judgement paradigm. It is a measure of generalized affective state rather than an immediate measure of welfare. Hence, this can be used in conjunction with other tools or independently to simply measure mood. It has the advantage that it can identify positive emotions and more practical species-specific measures that could then be used for other assessment tools. The technique has not been tested in reptiles to date, and only three studies have been performed on animals in a zoo setting, two of which were inconclusive [40]. 

Assessing the response of an animal to a novel environment is another approach to assessing animal welfare. It can demonstrate anxiety-like behaviors that will likely be species-specific and hence useful to incorporate as indicators in an assessment tool. Moszuti et al. (2017) investigated response to novelty in two species of reptile, the red-footed tortoise (*Chelonoidis carbonaria*) and the bearded dragon (*Pogona vitticeps*). The tortoises displayed an overall anxiety-like response when placed in a novel environment, and neck extension appeared to be a plausible sign of anxiety in this species. Conversely, while the bearded dragons did not display an anxiety-like response, they did seem to discriminate between familiar and novel environments [45]. This study suggests that novel testing may be very useful for assessing negative welfare and can be performed in reptiles. However, due to the vast diversity between species, more research needs to go into generalizing this approach across different taxa to ensure appropriate measures of welfare are being employed. 

## 5. Assessment of Negative Affective State

### 5.1. Biomarkers for Stress

Under baseline conditions, hormonal, energetic and immune systems are coordinated within individuals to maintain steady state function [46]. The effects of stressors dissociate this system and cause a multitude of physiological changes to the body. These result in measurable fluctuations in the concentrations of various analytes. 

#### 5.1.1. Sympathetic Adrenal Medullary (SAM) Response

Stress is a universal phenomenon that occurs in all vertebrates, and the effects of such are widely used for affective state assessment. The autonomic nervous system is activated as a short-term response to stress, leading to an immediate, direct and short-acting effect on most tissues in the body. Catecholamines are released from the adrenal glands in seconds and catabolized very quickly [47]. These include epinephrine (or adrenalin) which initiates the ‘fight or flight’ response in animals, endorsing short-term survival, and norepinephrine which is the primary neurotransmitter of the sympathetic nervous system responsible for changes in cardiovascular tone. The magnitude of the SAM response is reflected in the serum catecholamine concentration, blood glucose levels, heart rate and blood pressure [47]. 

Catecholamine levels in response to various stressors are poorly characterized in reptiles. In a study involving prolonged restraint of juvenile alligators out of water, the authors commented on the presence of elevated levels of both epinephrine and norepinephrine at initial restraint and bleeding, with norepinephrine remaining elevated for 24 h. Unexpectedly, epinephrine levels declined, despite the continued restraint stress [48]. Initial epinephrine levels reported in the alligators in this study were approximately twice those reported in an aquatic turtle, *Chrysemys picta*, submerged in anoxic water [49], and for the lizard, *Dipsosaurus dorsalis,* subject to exhaustive exercise [50]. Conversely, epinephrine values at restraint and initial sampling in the lizard, *Urosaurus ornatus*, were double those of the juvenile alligators [51]. These large differences in values probably reflects the influence of handling stress from blood sampling [48], which may vary with conditioning and between species, and highlights the difficulty in use of catecholamines for practical assessment of response to a stressor.

Glucose may also be used to assess stress in reptiles [46]. Hyperglycemia has been found in crocodilian and ophidians when exposed to stressors [47], due to the stimulation of gluconeogenesis and glycogenolysis by corticosterone [46]. Glucose was seen to increase fairly rapidly (within an hour) following capture for blood sampling in green turtles (*Chelonia mydas*) with elevated levels (suggestive of a hyperglycemia), present in healthy compared to diseased animals [52]. The trend in glucose levels with a peak at 4 h, followed by a decline to 24 h, was similar for both healthy and diseased animals. This implies that when interpreting glucose elevations as an indication of a stress response, consideration should be given to the health of the animal, perhaps due to the effects of disease on nutrient intake. 

Heart rate and blood pressure of various reptile species have been documented, but given the difficulty in documenting these parameters under field conditions and the influence of obtaining the measurements themselves on the stress response [47], they are not likely to be practical assessment tools. Heart rate variability has been investigated as a potential measure of welfare state in many mammalian species, by providing an indication of change in sympathovagal balance as a result of stressors, or temperament differences [53]. This method does not appear to have been investigated in reptiles for the purposes of animal welfare assessment, and yet may be a reliable and fairly non-invasive strategy. However, it is likely to be subject to influences such as external temperature or reptile metabolic activity and will therefore need detailed investigation considering these factors [54]. 

In conclusion, whilst measures of the SAM response have been traditionally used in animal welfare science to measure response to stressors, and are under-investigated in reptiles, further evaluation of these parameters is probably not a priority in reptile welfare research. They suffer from multiple confounding influences, are challenging to measure due to sampling method and really only provide a measure of immediate physiological response to stress, rather than sustained welfare state. Furthermore, these measures are difficult to interpret since: 1) they reflect only arousal of emotional response, being evoked by both positively and negatively valenced stimuli; and 2) it is challenging to factor in the many occurrences of these responses, often with no long-term detrimental effect, in assessment of overall welfare state based on the ‘sum’ of the animal’s individual experiences. 

#### 5.1.2. Hypothalamic Pituitary Adrenal (HPA) Axis

As a slightly longer-term response to stress, the Hypothalamic Pituitary Adrenal (HPA) axis initiates a hormonal cascade to cause the release of cortisol from the adrenal cortex [47]. Baseline cortisol levels maintain energy balances and are associated with seasonal and daily routines. Stress-induced cortisol promotes short-term survival but may also have negative effects on the body which will be discussed later [54]. It is widely accepted that measurement of products of the HPA axis will indicate that stress is being experienced by an animal. Most of this research is based on mammalian species. In reptiles, the main secreted glucocorticoid is corticosterone [47].

Physiologically, glucocorticoids have a multitude of effects on the body, but in the main cause lipolysis, fat redistribution, increased protein breakdown, suppressed immune function and gluconeogenesis. These effects are warranted to promote survival in short-term situations but will have dramatic consequences on the body over the long-term [47].

Corticosterone is influenced by a plethora of processes, including physiological (peaking when the reptile is most active) and environmental conditions [47]. For example, glucocorticoids have been shown to vary between individuals according to past experiences, geographical location, current energetic status, reproductive status, age and body condition [46]. 

Within captive reptiles there is evidence of an effect of group-housing on plasma corticosterone, with subordinate alligators having inherently higher plasma concentrations relative to their dominant conspecifics with higher growth rates [55]. Furthermore, wild counterparts have very different lives compared to those in captivity [22,56]. Dickens & Romero (2013) indicate that there is limited literature on how wild reptiles respond endocrinologically to stressors [57].

Gangloff et al. (2017) investigated two free-ranging populations of the common garter snake (*Thamnophis sirtalis*) and found that both plasma corticosterone and glucose concentration differed between georegions. It was suggested that the strategy or capacity to maintain function and energy balance with environmental variation may vary within a species [46]. Temperature can also affect absolute corticosterone concentration. Dupoué et al. (2013) reported an increase in corticosterone levels in a group of captive Children’s pythons (*Antaresia childreni*) exposed to low temperatures. The authors suggest that at a suboptimal temperature, an increase in corticosterone may help maintain an alert state essential for promoting immediate survival in these species. This study concluded that short-term changes in body temperature will affect absolute corticosterone concentration and should be considered whenever measuring this analyte in reptile species [58].

Baseline levels of corticosterone may differ between males and females during their reproductive cycle. Males tend to have a higher plasma corticosterone concentration, especially during times of increased activity and metabolism, such as spermatogenesis. This has been demonstrated in reptiles including desert tortoises (*Gopherus agassizii*) [59] and the six-lined racerunner lizard (*Cnemidophorus sexlineatus*) [60]. 

Corticosterone is typically measured invasively via a blood sample. However, less invasive sampling methods are available, such as fecal, urinary, salivary, tear, skin shed, and hair samples [22,61]. Due to the lack of baseline reference ranges in reptiles, interpretation is problematic [5]. Non-invasive corticosterone sampling methods in reptiles are understudied. However, a recent study was able to validate an enzyme immunoassay method for corticosterone detection in shed Komodo dragon skin. In this study, biological variation in corticosterone, as demonstrated by other researchers, was present. For example, corticosterone values were higher in males, juveniles and after spring skin shedding [61]. Further characterization of non-invasive measurement methods is an area ripe for future research, as these alternative methods are being more extensively used in mammalian species. 

There are other physiological measures, discussed briefly below, that may be useful; however, these may not be practical for routine welfare assessment. Many of these require blood sampling, itself a stressful procedure, as non-invasive techniques for sampling are not yet available. 

#### 5.1.3. Hematological and Biochemical Markers

When glucocorticoid hormones increase, white blood cells migrate between tissues and the circulation, leading to a decrease in circulating lymphocytes [62]. Increased corticosterone has been correlated to heterophilia and lymphopenia in some reptile species [47]. An increase in the ratio of heterophils to lymphocytes in whole blood is often seen in response to stress [46]. However, in Case et al.’s (2005) study on the eastern box turtle, whilst the enriched group displayed a significantly lower heterophil to lymphocyte ratio, there was no change in corticosterone concentration between the two groups [26]. Terrapins under hyperosmotic stress also tended to have increased hematocrit, heterophil count, sodium, chloride and potassium [47]. Whilst the heterophil to lymphocyte ratio has been widely investigated in mammals and birds, few examples of the use of this measure exist in reptilian species. 

Biochemical analysis of blood may reveal an increase in creatinine phosphokinase as part of a stress response [47], as observed in captured lace monitors (*Varanus varius*) [63]. A change in cholesterol and triglycerides can also occur, with these parameters tending to decrease, although this may be stressor dependent [47]. For instance, alligators experiencing cold shock demonstrated a significant increase in triglycerides and a decrease in cholesterol, whereas the same species experiencing restraint stress did not show a change in either lipid [48]. Hence, variability is not only seen between reptile species, but in response to the type of stress the reptile is experiencing, thus reducing the validity of these analytes. 

#### 5.1.4. Immunology

Measuring the antibody IgA has been suggested as an indicator of stress that could be used in zoo animals [18]. This protein can be measured non-invasively using feces and saliva. An acute stress response causes an increase of IgA immediately after the stressor, whereas a delayed stress effect tends to decrease IgA [64,65]. The mechanisms of such changes are only speculative and involve possible effects of psychosocial factors on this antibody’s transport system through the body [64]. Conversely, IgA has been shown to increase in situations of positive affective state in multiple studies; the mechanism behind this has not been explored [66]. IgA has been measured successfully in pig and dog saliva using absorbent cotton for collection [65,67]. It was found to be a more direct measure of stress than cortisol as it immediately returns to pre-stress levels when the acute stressor is removed [65]. IgA has also been useful in measuring stress in the rat, using filter paper discs to collect saliva [68], or through fecal analysis [69]. 

Immunoelectrophoresis is used to measure the IgA concentration, which is expensive and not easily accessible in facilities such as zoos. Based on the literature available, IgA does not appear to have been researched for use in reptiles. 

#### 5.1.5. Sampling Concerns

As alluded to previously, in developing practical and reliable animal-based measures of welfare state, methods need to be simple, easy to perform, repeatable and not inducive of excessive welfare compromise. The majority of the methods listed above require blood sampling. Options for blood sampling of reptiles include tail veins, the jugular vein, subcarapacial and occipitial sinuses, or cardiocentesis. The restraint required is often stressful for the reptile, and if not performed appropriately can result in other adverse effects, including physical injury. Risks to personnel when handling reptiles may include bites, scratches, envenomation and exposure to zoonotic pathogens.

The handling associated with blood sampling is generally a stressor for reptiles and hence the timing of blood sampling is essential for measuring many biomarkers. For example, capturing a captive reptile (species not stated) was shown to cause a measurable increase in corticosterone from anywhere between 3 min and 6 h [47]. Dupoué et al. (2013) managed this by bleeding pythons immediately, and taking additional samples 30 and 60 min later to account for variation [58]. Gangloff et al. (2017) collected a baseline blood sample within ten minutes to ensure corticosterone measures were not confounded due to handling stress [46]. Whilst these concerns are not unique to reptiles, knowledge of these confounding factors, and their latency of effect is needed when applying these biomarkers to welfare assessment. 

### 5.2. Behavioral Measures of Stress

Behavioral indicators of negative welfare in reptiles have not been well investigated in wild or captive reptiles. Warwick et al. (2013) refers to 30 behavior-based signs of stress in reptiles, as well as quiescence or comfort [5]. However, Warwick comments on the shortage of field-based observations in these species, and human-invoked responses that cause discrepancies.

Silvestre (2014) suggests that behavioral adaptation to stressors is the most effective response to reduce long-term consequences of exposure to stress-inducing conditions [47]. In captive reptiles that may be unable to escape stressors because of their environment, observation of their behavioral adaptations is expected to be a particularly useful indicator of welfare compromise. Useful behaviors indicative of chronic stress include aggression, anorexia, redirected activities, stereotypies and displacement behaviors. Stereotypies are considered detrimental when they make up 10% or more of waking behavior [34]. It has been postulated that stereotypies could be a coping mechanism, as they have been associated with the release of endorphins [22]. It is therefore controversial whether welfare at the time of performing the stereotypy is actually poor. However, the presence of a stereotypy can be used to infer a deficit in conditions, at some point in time, which caused the animal to adopt this behavior [18]. Maladaptive stereotypies such as the repetitive pacing seen in mammals, often attributed to stressors of captivity, have not been documented in reptiles. Reptiles may however perform repetitive behaviors, such as interaction with transparent boundaries or exploratory behavior [70]. Stressed lizards may also show a change in skin color, for example in bearded dragons (*Pogona vitticeps*) [71]. The chameleons (*Chamaeleonidae*) are well-known for their color changes. For instance, the dwarf chameleons (*Bradypodion transvaalense*) display behavioral changes, including color change, when faced with predators [72].

Sleep patterns are indicative of welfare, and in facilities such as zoos can be influenced by husbandry routines, proximity to the public, and the environment. However, there is a lack of research on the natural and healthy sleep patterns of many exotic animals [18]. Sleep is involved in an individual’s time budget, a disturbance in which is often used as an indicator of negative affective state [34]. 

In reptiles, time spent basking is an important consideration when evaluating behavioral time budgets, as time spent at the upper end of a species-appropriate temperature gradient is physiologically important [21]. Appropriate basking is a valid indicator of welfare for ectothermic animals, as it is closely related to appropriate thermal range, which is influenced by the animal’s evolutionary history [21]. Heat and light act in synchrony, and an excess or lack of either or both can result in pathological alterations [21], such as hypoactivity [5]. Basking may be an indicator of negative welfare states, since reptiles use behavior in modulating their response to both internal (physiological) and external conditions. For example, in response to ‘true’ fever or emotional fever caused through exposure to stressful events such as handling, reptiles preferentially seek warmer temperatures. Deficiencies in the environment such as low ambient temperatures, inadequate access to basking sites through competition or overcrowding, or the under-provision of other ‘cage facilities’ leading to ‘hyperbasking’ can all increase the time spent basking [5]. These signs represent animal-based measures reflecting the appropriateness of environmental inputs and should be included in welfare tools for reptiles.

## 6. Health as an Indicator of Welfare 

Physical health is a commonly used indicator of welfare, since minimizing disease and injury promotes comfort and functional capacity whilst reducing negative experiences such as pain, debility and weakness [73]. It is hypothesized that a positive affective state boosts immunity and improves physical health [13], supporting the use of health indicators in a welfare assessment tool. 

Warwick et al. (2013) describes common detrimental physical health signs in reptiles, including rostral abrasions, thermal burns, ventral dermatoses, and pica. Rostral abrasions, for instance, are mostly associated with abnormal repetitive behaviors [5], which can be a maladaptive or malfunctional sign of negative welfare [74]. Fine cuts on the snout of captive snakes are characteristic of repetitive behaviors [21]. Integument alterations or wounds are hence a valid indicator for a deficit in welfare which has been incorporated into several welfare assessment tools for mammals and birds. Stress has also been found to lead to egg binding in reptiles, a common issue in gravid lizards [21].

Chronic and progressive stress which leads to a prolonged increase in corticosterone results in muscle degeneration and growth suppression. This consequently predisposes to emaciation. However, some reptiles that are undergoing chronic stress will present with obesity [47]. Obesity also results from overfeeding in conjunction with reduced energy expenditure, which may be related to the captive environment and/or a failure to consider low reptilian metabolic rates, and high conversion efficiencies, when establishing feeding regimes [75]. Obesity has many detrimental effects, including dystocia and decreased lifespans. In reptiles, excessive fat can be stored in subcutaneous tissue or internal organs, causing derangements such as hepatic lipidosis, especially common in chelonians [9,76]. This supports the use of body condition scoring in a welfare assessment tool. It is important to consider however, that a body condition score system must be standardized in different species to optimize validity and minimize subjectivity.

A prolonged increase in corticosterone also results in immune depression and reproductive difficulty [47]. In comparison to mammals, reptiles have a weaker and slower humoral immune response. The response can be directly affected by the season, reproductive state and the environment, with immune function optimized within the species-specific preferred temperature range [9]. If a reptile’s immune response is compromised, there is a greater risk of infection [9]. Pees et al. (2010) found a significant correlation between poor husbandry conditions and presence of microorganisms in tracheal washes of captive Boidae [77].

Whilst welfare assessment across the lifespan of an animal is critical, special consideration should be given to welfare assessment in aged animals, which are at increased risk of negative experiences like pain. Reptiles generally have long lifespans [78]. The onset of age-related ailments may be insidious and only reflected in subtle reductions in performance and vitality [10]. Knowledge of the species of interest is of particular importance since there are considerable species differences in propensity to demonstrate age-related health effects; for example, testudines and crocodilians show almost negligible senescence, while many species of squamates show gradual senescence much like other vertebrates [79].

## 7. Welfare Assessment Tools

Draper & Harris 2012 reported that only 24% of 192 British zoos inspected met all legislated welfare standards [80]. Similar findings in other parts of the world might be expected. There are increasing efforts in many jurisdictions to ensure welfare standards are upheld; the Australasian Zoo and Aquarium Association now has animal welfare as the central focus of its accreditation program. In addition, several assessment tools have been developed over the last decade that have proved useful in monitoring welfare. The two welfare tools addressed in this paper were chosen since they were deemed most likely to be successfully translated to reptiles. These are the Five Domains Model and the Welfare Quality^®^ Protocol. 

### 7.1. Five Domains Model

The Five Domains Model is based on four physical, or functional domains, and one cumulative mental domain. Physical domains include nutrition, environment, health and behavior. Mental state represents the affective domain which results from the factors inherent in domains 1–4. This tool has the advantage of being readily adaptable to model species [6] and is simpler than the Welfare Quality^®^ Protocol, which requires more extensive grading and calculation. 

This tool has been extensively evaluated in a zoo environment in a study by Sherwen et al. (2018) which monitored its use in 339 species, including reptiles. Assessments took an average of 35–50 min per enclosure [1]. This is much longer than other adaptations, for instance taking only 2.5 min per individual when performed on extensively farmed ewes [81]. However, zoo enclosures must take groups into consideration, as well as interaction with all aspects of the enclosure.

The application of this tool in a zoo setting represents a prodigious attempt to monitor the change in welfare state over time across different species. However, it was suggested that when adapting the tool that it be used in conjunction with other tools to provide a more thorough evaluation of welfare, as it may be lacking in some respects [1]. The discussion of these elements is beyond the scope of this review. 

### 7.2. Welfare Quality^®^ Protocol

The Welfare Quality^®^ Protocol was designed to assess the welfare of intensively produced livestock using 12 criteria that reflect conditions meaningful to these animals, and being derived from four principles, much like the Five Domains Model [82]. The four welfare principles being good feeding, good housing, good health and appropriate behavior. Measurable indicators determined by expert consultancy were allocated to the criteria and were integrated in a step-wise fashion to obtain a final welfare assessment [82]. 

These protocols set out for pigs, poultry and dairy cattle are resource and time-intensive. For example, layer poultry assessment can take up to 7 h to complete [82], which hampers practical implementation of this tool. The Five Domains is superior in this respect.

The major limitation to this protocol is redundancy. That is, it was found through testing this tool in production animals that overall welfare classification could be explained by two resource-based measures only. A system based on these two measures could reduce the assessment time to a few minutes, though with the risk of losing a holistic view [83,84]. 

The Welfare Quality^®^ framework has been adapted to farmed fox and mink by Mononen et al. (2012), forming a benchmark for European fur farms [85]. Clegg et al. (2015) also adapted this framework to systematically assess the welfare of bottlenose dolphins (*Tursiops truncatus*), a common species displayed in zoos [34]. Similarly, Salas et al. (2018) adapted this framework to Dorcas gazelles (*Gazella dorcas*) in a zoo [86]. These adaptions have been successful in improving the welfare of these animals and are constantly being improved as they are trialled. 

Due to the abundance of research in the use of the Welfare Quality^®^ framework, the practical experiences of authors using this tool can be gleaned, and its associated benefits and drawbacks recognized. Hence, this tool was chosen here for adaptation to the pygmy blue-tongue skink.

### 7.3. Role of Zoo Keepers

Welfare assessment tools are very useful for keepers, who have become proxy informants for the animals under their care [18]. However, there is still much to learn about many species held in zoos; these gaps in knowledge limit the application of animal welfare tools to those species. Mehrkam & Dorey (2015) found that zoo keeper predictions of the most-preferred enrichments for animals in their care across several different taxonomic groups was the least accurate (5.8%) for reptiles [30]. Despite this, due to their knowledge, experience, and time spent with the animals they were evaluating, keepers should be best placed to recognize abnormal behavior, develop enrichment strategies, and assess the welfare of animals under their care [25]. Consequently, they must be equipped with the tools to formally perform these activities successfully. Therefore, when investigating the application of a tool to the pygmy blue-tongue skink, we incorporated feedback from keepers in both the design and piloting phase. 

## 8. Pygmy Blue-Tongue Welfare Assessment Tool

### 8.1. Development

The indicators explored for each criterion in the applied tool were chosen based on the limited literature written on reptile welfare, and through consultation with reptile and zoo veterinarians, and zoo keepers familiar with this species from Adelaide and Melbourne Zoos. The final application with explanatory notes and references is provided in Table 1. A short pilot study was performed at Adelaide Zoo by observing the lizards in their enclosures. Unlike the Welfare Quality^®^ Protocol, extensive grading was not used in the pilot study, since an exploratory approach was taken to develop this tool. The method of grading each indicator was adapted from that presented in Sherwen et al. (2018) [1], shown in Table 2. 

This investigatory assessment was performed from the viewpoint of a zoo visitor to reduce the potential effects of stress related to close proximity for observation and/or handling, and to accurately assess welfare whilst animals are on display [86]. Assessment was only conducted in winter, and assumptions of seasonal variation were made accordingly, in consultation with keepers. This only provides a ‘snapshot’ of welfare standards and may not fully reflect the influence of seasonal climate and breeding seasons on animal welfare [87]. Due to the ectothermic physiology of reptiles, and particularly the cryptic nature of this species, many of the indicators chosen are near impossible to assess during winter, when the lizards are dormant in their burrow. This has been a major limitation in developing criteria for welfare assessment in this species, and is likely an issue in many other reptiles that also exhibit this level of seasonal inactivity. Similarly, many reptiles are nocturnal or crepuscular and hence assessment may be compromised if performed diurnally. 

Measurement of heart rate and blood pressure are not included in this application and are likewise suggested to be excluded from a developed tool. Auscultation of the heart of a reptile is problematic. There is friction between the stethoscope and the scales, and generally only low amplitude heart sounds can be heard. Doppler monitors are a useful alternative for the assessment of heart rate and rhythm, and subjective assessment of blood pressure, in reptiles. Physical contact with the reptile is still required, and availability may be a limiting factor. Peripheral blood pressure may be measured with surgically placed intra-arterial devices in anaesthetized reptiles. There is currently no effective indirect blood pressure method described in these species [9]. 

Throughout the application of this protocol, one animal-based measure initially proposed was omitted. Tail autotomy is a sign correlated with fear in reptiles [5]. However, despite autotomy planes being well developed in *Scincidae*, the genus *Egernia* and *Tiliqua* have reduced planes, and are therefore unlikely to drop their tails. Hence this measure was excluded [88].

### 8.2. Trial

The pilot study performed took place at Adelaide Zoo, taking two hours to complete. The individuals observed were a breeding pair. Three resource-based indicators were added as criteria, following observations and discussion with keepers. These were enclosure cleanliness, enclosure maintenance and group size. The method of grading each indicator was adapted from Sherwen et al. (2018) [1], shown in Table 2. This method was chosen due to its simplicity, which may aid in reducing subjectivity of grading. The overall welfare score was determined by averaging the grades out of the maximum score possible, the highest score possible being two for each indicator. It is suggested that more extensive grading systems be investigated when developing a welfare assessment tool for reptiles, to allow for improved sensitivity of the tool and consequently more accurate monitoring of welfare over time. 

A number of criteria could not be assessed due to gaps in opportunities. One of the subjects was not observed drinking, likely due to wariness. Anorexia and hunting behavior could not be assessed, since food was not presented. This highlights the need for good record keeping, since ‘food intake’ was able to be scored due to the notes taken two days prior on feeding day, where both subjects ate the food presented to them. It is suggested from these findings that assessment be completed over at least two days, one of which being a scheduled feeding day for the zoo staff. Alternatively, keeper record keeping could be used as a means of formally documenting behaviors relating to particular criteria over time, allowing keeper records to be used to inform the welfare assessment.

During the pilot study, seven animal-based measures and three resource-based measures received a score of lower than two. Scores for the indicators ‘basking’, ‘wounds’, ‘respiratory system’, ‘co-occupant aggression’ and ‘exploratory behavior’ differed between the individuals of this breeding pair. This highlights the need for independent scoring of animal-based measures due to an individual’s past experiences and biological variation. It also highlights the importance of documenting the interaction between individuals in the same space. The thermal range of the enclosure was identified to be suboptimal for the species and was corrected as a result. By documenting the interaction of this pair with enclosure furnishings, the value of these items may be gauged in order to develop an enrichment protocol for this species. Enrichment could be improved by varying the furnishings more often and documenting their use to monitor value. However, it was observed that the enclosure was sufficiently complex so as to provide escape opportunities for the individual skinks when threatened. The lower score for the ‘capture and restraint’ method and ‘cloacal evacuation’ might suggest that the lizards would benefit from being conditioned to handling. However, handling is not a frequent event for these lizards, and for a species being maintained as an assurance population to support field conservation efforts, humanization may be detrimental to survival following a future release to the wild. 

The total possible score for this mating pair was calculated as 138 (the highest score for each indicator being two for each indicator and each individual, except for the resource-based indicators which were allocated a maximum of two for the pair). When counting the unknown scores as zero, the score obtained in this pilot was 109.5 out of 138 or 79%. The welfare threshold used in Sherwen et al. (2018) was 60%, which has been used in our application in conjunction with the grading system [1]. It is proposed that welfare thresholds be individualized to a zoo or institution based on respective goals and past experiences, and should be decided through discussion with zoo keepers, veterinarians and other experts in the field. Utilizing the 60% threshold in this exploratory application of the criteria, this mating pair were assessed as having good welfare.

## 9. Conclusions

In conclusion, we have proposed a modification of the EU Welfare Quality^®^ framework to a species of reptile with the use of predominantly (72.5%) animal-based measures. Due to the quiet nature of the pygmy blue-tongued skink, a variety of measures were needed to obtain a holistic view of welfare. These included behavioral, health and husbandry indicators. This application requires further development of the criteria with consideration of species-specific behaviors and requirements, and validation. However, we advance that the proposed measures provide a foundation for future development of a welfare assessment tool for reptiles. It should be noted that a single pilot study using crude measurements is not accurately indicative of a welfare score, and this tool should be trialled further with more extensive grading methods. Ultimately, a developed tool will aid in ascertaining, and therefore improving, welfare for the pygmy blue-tongue skink and other reptile species. Possible areas of further research include the application of the Five Domains model to a reptile species, and the customization of the applied tool to even less behaviorally expressive species of reptile, such as some snake species. Experimental applications of welfare assessment should also be investigated, such as cognitive bias testing, to determine its efficacy in reptiles. The utilization of IgA as a non-invasive measure of stress should also be considered, as well as extensive enrichment preference testing in reptiles. As human caretakers of animals we are obligated to strive for improved animal welfare, and it is a priority of zoos to promote good welfare.

## Figures and Tables

**Table 1 animals-09-00027-t001:** Application of the welfare assessment tool of the pygmy blue-tongue skink (PBTS, *Tiliqua adelaidensis*) based on the Welfare Quality^®^ framework.

Category	Welfare Quality^®^ Criterion	Welfare Quality^®^ Measure
Good Feeding	Absence of prolonged hunger	**Body condition score**—*utilizing a standardized system, e.g., using the Zoo Information Management Software utilized by zoos globally* [89]. **Persistent food-seeking behavior**—*indicating food restriction [73]. If the PBTS is fully out of burrow seeking food, replacing normal behaviors such as basking, relaxed exploration, etc.* **Food intake**—*observed relaxed when eating, or recording food left, indicating the reptile is eating when given the appropriate diet.*
	Appropriate diet	**Appropriate diet provided**—*to provide satiety [73]. The PBTS is omnivorous, and their diet should consist of 50% vegetables, 25% fruits, and 25% invertebrates such as snails [9].*
	Absence of prolonged thirst	**Signs of dehydration**—*sunken eyes and/or multiple skin folds [90,91].* **Unhurried drinking**—*observed relaxed drinking as a normal maintenance behavior [5].* **Cleanliness and (true) availability of water**—*appropriate quality of water [22], replaced at an appropriate frequency, provided in species-appropriate presentation. Variation in methods of provision, e.g., spraying vegetation, regular bathing.*
Good Housing	Comfort around resting	**Animal cleanliness**—*e.g., fecal staining. Scat piling is a behavior performed by the PBTS [92] and may be disturbed when experiencing a deficit in welfare, resulting in fecal staining or random scatting [93,94].* **Enclosure cleanliness**—*Unhygienic conditions can lead to disease and stress [9]. Is the remainder of the enclosure cleaned regularly? Scat piles should be removed regularly only during brumation to prevent parasitic loading (scats serve as an important mate attractant signal) [11,93,94].*
	Thermal comfort	**Basking (time budget)**—*presents as relaxed and stretching out [5] with the head orientated to the heat source [95]. PBTSs will come out of (but stay close to) their burrow to bask in the morning, retreat midday and come out again in the late afternoon.* Behavioral indicators that may be, in part, indicative of inappropriate thermal range [5]: - **Hypoactivity**: *unalert; abnormal low-level physical activity.* - **Anorexia**: *refusal to eat available and appropriate feed.* - **Open-mouth breathing**: *sporadic, slow, open-mouth respiration or gasping.* - **Panting**: *rapid open-mouth breathing.* - **Ease of shedding and dermatophagia**: *shedding is a regular, healthy maintenance behavior of reptiles [5] and depends on the environment, diet and hydration of reptiles [90]. The skin sheds in stages and comes off flexible and transparent, retained dry/brown skin should not be present. This species naturally consumes its own shed skin [96].* **Appropriate thermal range**—*thermal gradient of 28–32° is preferred by this species [9]* **Enclosure maintenance**—*adequate records, monitoring and maintenance of heat and UVB lights to ensure the appropriate thermal and spectral range is being provided [9].*
	Ease of movement	A restrictive, deficient or inappropriate environment can result in stress-induced behaviors such as [5]: - **Hyperactivity**: *abnormal high-level physical or redundant activity.* - **Flattened body posture**: *flattening of body against a surface, sometimes associated with hyper-alertness.* - **Head-hiding**: *deliberate hiding of head including under objects/substrate.* **Enclosure size**—*minimal requirements must be met to ensure movement ease [81,85,86]. The pygmy skink is approximately 90 mm snout-to-vent in length [97]. Hence, it requires a minimum of 0.2 m^2^ enclosure space according to [90], or a minimum length of 100 cm by [98].*
	Interaction with burrows	**Movement into burrow**—*the PBTS moves head-first into a natural burrow, but exhibits a reversing behavior to enter an artificial burrow due to the lack an expanded internal chamber [99].* **Atypical locations**—*outside burrow in Winter or inside burrow and not basking in Summer (except for sleep or escaping a perceived threat) is atypical.* **Availability and provision of burrow**—*PBTS naturally inhabits spider burrows in grassland sites in South Australia [8]. Burrow at least 12 cm deep [100]. Burrows made of natural material, of appropriate length (juvenile 20 cm, adult 30 cm) and diameter (12–18 mm).*
Good Health	Absence of injuries	**Integument alterations (wounds)**—*absence of wounds promotes good health and functional capacity [73]. Wounds may be sustained during courtship behavior. Wounds related to interaction with the enclosure, such as abnormal repetitive behaviors resulting in rostral lesions, are of concern [5].* **Lameness**—*abnormal gait or stance, attributable to e.g., injury, inappropriate substrate [22] or metabolic bone disease [9].*
	Absence of disease	**Musculoskeletal system**—*deformities/weakness from e.g., metabolic bone disease, commonly afflicting lizards and turtles in captivity [9].* **Respiratory system**—*ocular/nasal discharge, dyspnea. May reflect inappropriate hygiene, climatic conditions or inappropriately dusty substrate [9].* **Stomatitis**—*ginigival bleeding, tooth loss. Typically associated with immunosuppression (secondary to poor husbandry) [9].* **Parasites**—*externally visible ticks/mites in skin folds, nostrils, corneal rims, and infraorbital pits [90]. Internal parasites may be associated with gastrointestinal signs, or weight loss.*
	Absence of pain induced by management procedures	**Access to veterinary treatment when necessary**
Appropriate behavior	Expression of social behaviors	**Affiliative behavior**—*promotes playfulness, maternal reward and sexual gratification [73]. In PBTSs, the dam shares the burrow with neonates for short time after birth—juveniles leave and inhabit burrows close to site and move away as they grow. Male removed as juveniles roam to prevent aggression.* **Co-occupant aggression**—*overly aggressive or defensive displays, biting, chasing cage mate, potentially causing functional impairment, can be associated with courtship routines, lack of space in the enclosure or hunger [5]. In PBTSs, aggression is often displayed by the female [101].* **Group size**—*inappropriate grouping of animals leads to pathology [22]. Should be housed singularly, as a mating pair or as a dam and her offspring. Males are not to be housed together. A minimum of 3 burrows per lizard is required for PBTS.*
	Expression of other behaviors	**Hunting**—*voluntary hunting supports excitation and reward [73]. Attempts to follow/lunge at invertebrates with success.* **Interaction with transparent boundaries**—*persistent attempts to push against, crawl up, dig under or pace the transparent barriers of enclosure demonstrates a disturbance in welfare [5].* **Enrichment program**—*documented interaction with, safety of, and use of the data of the program [34]. Complexity of enclosure.*
	Good human-animal relationships	Signs of human-directed aggression [5]: - **Hissing and/or biting**: *sign of fear (urination, defecation, excretion of substance from cloaca when handled) [5], mock/real strikes using jaws or tail, hissing with inflation and deflation of the body.* - **Cloacal evacuation.** **Capture/restraint technique and conditioning**—*does the method minimize stress?*
	Positive emotional state	**Exploratory behavior**—*promotes positive experiences of engagement and control [73]. Relaxed interest/awareness in novel objects and furnishings in enclosure; calmly smelling or tasting objects or air with tongue-flicking; unhurried body movements and locomotion; head ‘peeping’ out of burrow in a relaxed manner throughout the day.*

**Table 2 animals-09-00027-t002:** Grading system reproduced from Sherwen et al. (2018) [1]. Overall welfare score determined by summation and determination of percent out of the maximum score possible. Welfare threshold to be aimed for is 60% as suggested by Sherwen et al. (2018).

Level	Score	Description
Resource-Based Risk Level	0	High risk: e.g., resource considered to be inadequate for animal and likely to have welfare implications.
	1	Moderate risk: e.g., resource considered to be suboptimal and improvements needed.
	2	No observable risk: e.g., resource provision considered to be good and species-appropriate according to natural behavioral biology.
Animal-Based Welfare Level	0	Poor: e.g., animal either under or over weight; behavioral abnormality present; limited behavioral diversity observed compared to that expected for the species; shows little engagement with and is excessively fearful of keepers.
	1	Moderate: e.g., animals slightly over or under weight; have observed signs of behavioral abnormality but not frequent; displays limited behavioral repertoire; somewhat engaged with environment and keepers.
	2	Good: e.g., animals in good condition; no signs of behavioral abnormality; displays high levels of behavioral diversity as expected for the species; appears engaged in environment and with keepers.
Unknown	-	Team considers they do not have information critical to make a judgement.

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
