# Peer review of "A Review of Welfare Assessment Methods in Reptiles, and Preliminary Application of the Welfare Quality® Protocol to the Pygmy Blue-Tongue Skink, Tiliqua adelaidensis, Using Animal-Based Measures"

_animals, 2019, doi:10.3390/ani9010027_

Round 1
Reviewer 1 Report
This document presents a review on welfare assessment methods and also presents a protocol for the evaluation of welfare in a reptile, the pygmy blue-tongue skink. This is a novel approach because there is no existing welfare protocol for that species. The review is easy to follow, however the protocol for the pygmy blue-tongue skink itself could be improved.
It is a review paper, however, I am not sure if the protocol can be presented as a review, as it is developed for the first time in this study. This protocol has not been applied on the species or at least the results are not presented in this review, which I consider not sufficient. Since the results of the protocol are not presented, there is not a discussion or well-driven conclusions, nor a possibility for the authors to discuss improvements of the protocol.
L34: Welfare Quality, without EU
L47: There is a double space between [2]. and As.
L154: ‘it is virtually impossible to create an acceptable copy of the animal's natural habitat’. The word ‘acceptable’ can be very controversial, I suggest deleting this word in this sentence.
L172-174: ‘For instance, affiliative behavior between social species is a useful indicator for amateur welfare assessors, however it is reported that it may not necessarily be reflective of positive affective states’. This sentence is not true, as the observation of affiliative behaviours is a broadly used welfare indicator by researchers as well as experts on the field.
L 183: The words ‘being obvious’ should be deleted, as it is not obvious in all the cases.
L188-189: Maybe it would be better to rephrase the sentence. Reptiles have less advance facial musculature compared with mammals and this difficult the use of facial expression as indicator in reptiles.
L201: There is a double space between [8]. and Burghardt.
L197-210: These two paragraphs are confusing, as it is stated that ‘play’ has not been documented until recently in reptiles, but at L206 it is said that there have been many documented play behaviors in turtles but only one source is mentioned.
L229-230: This statement is controversial, as it has not been proved that ‘Stress is a universal phenomenon that occurs in all living animals, however primitive they may be’.
L235: There is a double space between tone. and The.
L303: I think it might be interesting for your paper to add the reference of a recent paper about the use of shed skin of Komodo dragons to assess corticosterone: Annaïs Carbajal, Oriol Tallo-Parra, Laura Monclús, Manel Aresté, Hugo Fernández-Bellon, Vanessa Almagro and Manel Lopez-Bejar (2018) Corticosterone measurement in Komodo dragon shed skin. Herpetological Journal 28: 110-116
L425: General comment about this section: being this a review I found that more information could be added regarding the current available welfare assessment tools.
L432-433: Which are these domains?
L448: Which are the four principles?
L458-462: Two protocols for wild animals in captivity that were based in the Welfare Quality Protocols are included, but there is also a recent one that should be included in this paragraph and that it is used for references later on the paper: Marina Salas, Xavier Manteca, Teresa Abáigar, Maria Delclaux, Conrad Enseñat, Eva Martínez-Nevado, Miguel Ángel Quevedo, Hugo Fernández-Bellon (2018) Using farm animal welfare protocols as a base to assess the welfare of wild animals in captivity - Case study: dorcas gazelles (Gazella dorcas). Animals 8: 111
L235: There is a double space between Zoo. and The.
L477-484: Sherwen et al (2018) use the Five Factor as a base, while the authors of this study use the Welfare Quality Project. How was the fusion of these two methods done?
L507-520: Where is the score system explained? Welfare Quality does not use a system of summing scores. Which is the reason why 60% is considered enough for good welfare? Which were the lower scored measures after the application of the protocol? Any conclusions?
Table 1: On the description it is said that animal-based measures will be formatted in bold, but all the measures are in bold, including resource-based measures.
It is not clear in all the cases which justification pairs with each measure. Also, the sources are not clear to which justification are referring.
Some of the indicators need better explanation on how to apply them. Examples:
· Body condition score: which system is used? Why? It is just a subjective opinion of the observer or are there pictures of what is score 1 to 9 for this species?
· Cleanliness of the animals: is it clean vs not clean? What is clean? What is dirty?
· How long should the behavioural observations be? How many sessions?
· Enclosure size requirement of 0.2m2? Based on what?
· Exploratory behaviour can be added in ‘positive emotional state’? Are we measuring the emotions of the skinks by using this behavioural indicator?
These are some examples, but I think most of the indicators lack an explanation on how to apply them, making the tool difficult to use (unfortunately) and to replicate it in other settings.
‘Access to vet treatment’ should be ‘access to veterinarian treatment’.
What are transparent boundaries? The definition is not self-explanatory.
I lack a clarification on why why the 12 criteria of Welfare Quality are increased to 14 criteria for the pygmy blue-tongue skink.
Table 2: Again, which is the reason why 60% is considered enough for good welfare?
The reference needs to be added in the format of the journal.
L530: I miss the specific results of the application with a discussion. I understand that this is presented as a review paper, but the presence of the protocol makes arguable that this is ‘just’ a review.
Author Response
Comment 1
It is a review paper, however, I am not sure if the protocol can be presented as a review, as it is developed for the first time in this study. This protocol has not been applied on the species or at least the results are not presented in this review, which I consider not sufficient. Since the results of the protocol are not presented, there is not a discussion or well-driven conclusions, nor a possibility for the authors to discuss improvements of the protocol.
Response 1
We understand the reviewer’s concern in combining a review with an assessment tool. We would like to preserve the tool in the paper as whilst these measures might seem obvious to reptile experts, we believe that for amateur keepers they are not self-evident. We therefore feel that our proposed assessment indicators could guide future research into and validation of a tool. We appreciate that the tool is at an early stage. In light of the reviewer’s comments we have attempted to highlight throughout the paper that this is a pilot. We have also reduced our referral to our indicators as a ‘tool’ since this implies a developed method. Changes have been made in multiple sections as described below. We have shortened the tool presented to demonstrate how we think a basic adaptation of the Welfare Quality Tool would look in this species. Given that we haven’t fully investigated the tool and extensively trialled it we determined it might be more appropriate to present a comprehensive paper on this at a later date.
The title has been amended to, ‘A review on current welfare assessment methods in reptiles, and preliminary application of the Welfare Quality® Protocol to the pygmy blue-tongue skink, Tiliqua adelaidensis, using animal-based measures’.
Lines 19-21 have been rewritten to ‘We also explore the application of the Welfare Quality® Protocol to the pygmy blue-tongue skink, an endangered species. This application presents examples of predominantly animal-based indicators that may be further investigated for use in a tool in this and other reptile species’
Lines 30-32 have been revised to, ‘Finally, we examine some widely-used welfare assessment tools in mammals and explore the application of the Welfare Quality® Protocol to the endangered pygmy blue-tongue skink, Tiliqua adelaidensis. We propose that this framework can form the basis for the development of taxon-specific tools with consideration of species-specific biology.’
Lines 59-60 have been revised to, ‘Finally, we explore the application of the Welfare Quality® Protocol to the pygmy blue-tongue skink (Tiliqua adelaidensis), an endangered species of reptile’
Lines 122-124 have been revised to, ‘Animal-based measures are also predominantly featured in the proposed welfare indicators for the pygmy skink.’
Lines 571-573 have been revised to, ‘Therefore, when investigating the application of a tool to the pygmy blue-tongue skink, we incorporated feedback from keepers in both the design and piloting phase.’
Lines 577-579 have been revised to ‘The indicators explored for each criterion in the applied tool were chosen based on the limited literature written on reptile welfare, and through consultation with reptile and zoo veterinarians, and zoo keepers familiar with this species from Adelaide and Melbourne Zoos.’
Lines 585-587 have been amended to, ‘This investigatory assessment was performed from the viewpoint of a zoo visitor to reduce the effects of stress related to handling, and to accurately assess welfare while animals are on display…
Line 596-597 now reads, ‘Measurement of heart rate and blood pressure are not included in this application and is likewise suggested to be excluded from a developed tool’.
Line 604 reads ‘Throughout the application of this protocol….’
Lines 685-695 now reads, ‘In conclusion, we have proposed a modification of the EU Welfare Quality® framework to a species of reptile with the use of predominantly (72.5%) animal-based measures. Due to the quiet nature of the pygmy skink, a variety of measures were needed to obtain a holistic view of welfare. These included behavioral, health and husbandry indicators. This application requires further development of the criteria with consideration of species-specific behaviors and requirements, and validation. However, we advance that the proposed measures provide a foundation for future development of a welfare assessment tool for reptiles. It should be noted that a single pilot study using crude measurements is not accurately indicative of a welfare score, and this tool should be trialled further with more extensive grading methods. Ultimately, a developed tool will aid in ascertaining and therefore improving welfare for the pygmy blue-tongue skink, and other reptile species.’
Comment 2
L34: Welfare Quality, without EU
Response 2
EU has been removed at line 34.
Comment 3
L47: There is a double space between [2]. and As.
Response 3
The extra space has been removed.
Comment 4
L154: ‘it is virtually impossible to create an acceptable copy of the animal's natural habitat’. The word ‘acceptable’ can be very controversial, I suggest deleting this word in this sentence.
Response 4
We understand the reviewer’s concern and this now reads ‘In practice it is virtually impossible to comprehensively replicate in captivity the animal's natural habitat’ at 174-175.
Comment 5
L172-174: ‘For instance, affiliative behavior between social species is a useful indicator for amateur welfare assessors, however it is reported that it may not necessarily be reflective of positive affective states’. This sentence is not true, as the observation of affiliative behaviours is a broadly used welfare indicator by researchers as well as experts on the field.
Response 5
Lines 203-205 have been rewritten as, ‘Affiliative behavior between individuals of a social species can be readily observed, making this a useful indicator for amateur as well as experienced welfare assessors.
Comment 6
L 183: The words ‘being obvious’ should be deleted, as it is not obvious in all the cases.
Response 6
The words ‘being obvious’ have been deleted and the sentence now reads ‘Vocalization is also a commonly used indicator of affective state which can be measured non-invasively’
Comment 7
L188-189: Maybe it would be better to rephrase the sentence. Reptiles have less advance facial musculature compared with mammals and this difficult the use of facial expression as indicator in reptiles.
Response 7
We apologise for this error in proof-reading. This sentence now reads, L217-218 ‘Reptiles have less advanced facial musculature making it challenging to utilize facial expression as an assessment tool’
Comment 8
L201: There is a double space between [8]. and Burghardt.
Response 8
This has now been amended.
Comment 9
L197-210: These two paragraphs are confusing, as it is stated that ‘play’ has not been documented until recently in reptiles, but at L206 it is said that there have been many documented play behaviors in turtles but only one source is mentioned.
Response 9
We apologise for the contradiction. Line 208-209 stating that ‘Much like the concept of advanced parental care, play has not been documented in reptiles until quite recently’ has been removed. Lines 239-240 has been changed to ‘There have been many documented play behaviors in turtles (see [12] for summary).’
Comment 10
L229-230: This statement is controversial, as it has not been proved that ‘Stress is a universal phenomenon that occurs in all living animals, however primitive they may be’.
Response 10
We agree that this statement is controversial and consequently lines 277-278 have been rewritten as, ‘Stress is a universal phenomenon that occurs in all living vertebrates, and the effects of such are widely used for affective state assessment’.
Comment 11
L235: There is a double space between tone. and The.
Response 11
This has now been amended.
Comment 12
L303: I think it might be interesting for your paper to add the reference of a recent paper about the use of shed skin of Komodo dragons to assess corticosterone: Annaïs Carbajal, Oriol Tallo-Parra, Laura Monclús, Manel Aresté, Hugo Fernández-Bellon, Vanessa Almagro and Manel Lopez-Bejar (2018) Corticosterone measurement in Komodo dragon shed skin. Herpetological Journal 28: 110-116
Response 12
We thank the reviewer for this useful reference and have included referral to it at lines 365-369 in the revised manuscript.
Comment 13
L432-433: Which are these domains?
Response 13
An expansion of this section with the domains listed has been included at lines 523-524
Comment 14
L448: Which are the four principles?
Response 14
The four principles have now been listed at lines 541-542 of the revised manuscript.
Comment 15
L458-462: Two protocols for wild animals in captivity that were based in the Welfare Quality Protocols are included, but there is also a recent one that should be included in this paragraph and that it is used for references later on the paper: Marina Salas, Xavier Manteca, Teresa Abáigar, Maria Delclaux, Conrad Enseñat, Eva Martínez-Nevado, Miguel Ángel Quevedo, Hugo Fernández-Bellon (2018) Using farm animal welfare protocols as a base to assess the welfare of wild animals in captivity - Case study: dorcas gazelles (Gazella dorcas). Animals 8: 111
Response 15
We thank you for the suggestion, we have revised the paragraph at line 555 to incorporate this suggestion.
Comment 16
L235: There is a double space between Zoo. and The.
Response 16
This has been amended in the revised manuscript.
Comment 17
L477-484: Sherwen et al (2018) use the Five Factor as a base, while the authors of this study use the Welfare Quality Project. How was the fusion of these two methods done?
Response 17
The Welfare Quality Protocol was chosen to be applied to the pygmy skink in this paper due to its more abundant application in other captive animals (the farmed mink/fox, dolphins and gazelles as briefly discussed at lines 503-505) compared to the Five Freedoms model. It was decided that the grading system for the Welfare Quality® Protocol was too extensive for a pilot study such as the one done in this paper, and hence the grading system used by Sherwen et al was instead utilised. We mention in the conclusion at lines 691-693 that these were crude measurements and a more elaborate grading method should be used when developing a tool. That is, we are not trying to fuse these methods, but Sherwen et al had a simpler grading system that could be applied to the suggested indicators for use in the pilot study. This was due to lack of time and computerised software. Since this pilot study is crude, we only wish to present it as a stepping stone to much more extensive research (see Response 1).
Comment 18
L507-520: Where is the score system explained? Welfare Quality does not use a system of summing scores. Which is the reason why 60% is considered enough for good welfare? Which were the lower scored measures after the application of the protocol? Any conclusions?
Response 18
We understand that the grading system has not been well explained. As mentioned in Response 17, we used the simple system of averaging scores used in Sherwen et al due to lack of time/resources. We have added into lines 613-619 an attempt to amend this, ‘The method of grading each indicator was adapted from that presented in Sherwen et al (2018) [1], shown in Table 2. This method was chosen for the pilot study due to its simplicity, which may aid in reducing subjectivity when it comes to grading each of the indicators. The overall welfare score is determined by averaging the grades out of the maximum score possible, the highest score possible being two for each indicator. It is suggested that more extensive grading systems be investigated when developing a welfare assessment tool for reptiles, to produce statistics and allow more accurate monitoring of welfare over time.’
Likewise, the threshold of 60% is used simply as a benchmark for this pilot study (reasons for Sherwen using 60% are described in their study). Welfare thresholds should likely be individualized to a zoo/institution based on their respective goals and past experiences. We have added lines 648-653 to explain this, ‘The welfare threshold used in Sherwen et al (2018) was 60%, which has been used in our application in conjunction with the grading system [1]. It is proposed that welfare thresholds be individualized to a zoo or institution based on respective goals and past experiences, and should be decided through discussion with zoo keepers, veterinarians and other experts in the field. Utilizing the 60% threshold in this exploratory application of the criteria, this breeding pair were assumed to have good welfare.’
Scores lower than 2 included: ‘basking’ as the female of the pair basked but retreated into her burrow with human movement and loud sounds; ‘thermal range’ as the recorded temperature range for that day was lower than recommended for this species; ‘movement into burrow’ as behaviour suggested a lack of an internal chamber in the burrow; ‘wounds’ as the male had a slight excoriation on the dorsal surface of his head; ‘respiratory system’ as male was observed to sneeze but infrequently; ‘co-occupant aggression’ as female (instigator) approached male (subject), who retreated into burrow with defensive display of hissing/gaping (this was only observed once with no physical contact and there were opportunities for escape); ‘enrichment’ as enclosure furnishings are only added to/changed twice a year and its use is not documented; ‘cloacal evacuation’ as handling was not observed but personal communication with keeper suggests this is common; ‘capture/restraint’ as low stress method is used (not observed) but some aggression/cloacal evacuation exhibited (personal communication with keeper); ‘exploratory behaviour’ for the female as she showed less behavioural repertoire and was only somewhat engaged in environment. We have added some discussion on this at lines 630-632.
We have some conclusions that have been added as lines 629-644, ‘During the pilot study, seven animal-based measures and three resource-based measures received a score of lower than two. Scores for the indicators ‘basking’, ‘wounds’, ‘respiratory system’, ‘co-occupant aggression’ and ‘exploratory behavior’ differed between the individuals of this breeding pair. This highlights the need for independent scoring of animal-based measures due to an individual’s past experiences and biological variation. It also highlights the importance of documenting the interaction between individuals in the same space. The thermal range of the enclosure was identified to be suboptimal for the species and was corrected as a result. By documenting the interaction of this pair with enclosure furnishings, the value of these items may be gauged in order to develop an enrichment protocol for this species. Enrichment could be improved by varying the furnishings more often and documenting their use to monitor value. However, it was observed that the enclosure was sufficiently complex so as to provide escape opportunities for the individual skinks when threatened. The lower score for the ‘capture and restraint’ method and ‘cloacal evacuation’ might suggest that the lizards would benefit from being conditioned to handling. However, handling is not a frequent event for these lizards, and for a species being maintained as an assurance population to support field conservation efforts, humanization may be detrimental to survival following a future release to the wild. ‘
Comment 19
Table 1: On the description it is said that animal-based measures will be formatted in bold, but all the measures are in bold, including resource-based measures.
Response 19
We have removed reference to this and have just used highlighting to aid in reading the table not for discriminating between animal and resource-based measures.
Comment 20
It is not clear in all the cases which justification pairs with each measure. Also, the sources are not clear to which justification are referring.
Response 20
We apologise for this lack of clarity. We believe the simplification of the welfare Quality adaptation and its presentation in the Table here aids in clarity. References are now clearly aligned.
Comment 21
Some of the indicators need better explanation on how to apply them. Examples:
· Body condition score: which system is used? Why? It is just a subjective opinion of the observer or are there pictures of what is score 1 to 9 for this species?
· Cleanliness of the animals: is it clean vs not clean? What is clean? What is dirty?
· How long should the behavioural observations be? How many sessions?
· Enclosure size requirement of 0.2m2? Based on what?
· Exploratory behaviour can be added in ‘positive emotional state’? Are we measuring the emotions of the skinks by using this behavioural indicator?
These are some examples, but I think most of the indicators lack an explanation on how to apply them, making the tool difficult to use (unfortunately) and to replicate it in other settings.
‘Access to vet treatment’ should be ‘access to veterinarian treatment’.
What are transparent boundaries? The definition is not self-explanatory.
I lack a clarification on why the 12 criteria of Welfare Quality are increased to 14 criteria for the pygmy blue-tongue skink.
Table 2: Again, which is the reason why 60% is considered enough for good welfare?
The reference needs to be added in the format of the journal.
L530: I miss the specific results of the application with a discussion. I understand that this is presented as a review paper, but the presence of the protocol makes arguable that this is ‘just’ a review.
It is a review paper, however, I am not sure if the protocol can be presented as a review, as it is developed for the first time in this study. This protocol has not been applied on the species or at least the results are not presented in this review, which I consider not sufficient. Since the results of the protocol are not presented, there is not a discussion or well-driven conclusions, nor a possibility for the authors to discuss improvements of the protocol.
Response 21
As described above we have modified the table extensively to present a basic adaptation of the Welfare Quality framework to this species. This is designed to show the reader how we believe this can be done but is not intended to provide any great detail on this, since we have not fully investigated and trialled the adaptation.
The enclosure size requirement is based on a veterinary reference. We hope by amending the alignment of the references and justifications of the table, this will be clear.
As discussed, exploratory behaviour has been described as an indicator of valence of affective state, and has been demonstrated in reptiles. Without entering the vast area of cognition, we believe this behaviour is a valuable indicator for positive emotional state.
We hope changes made mentioned in Response 18 have amended the 60% threshold comment.
We also hope Response 1 and 18 have amended comments made on the results of the pilot study and in relation to the protocol being referred to as part of the review.
Reviewer 2 Report
Manuscript ID: animals-392714
A review on current welfare assessment methods in reptiles, and the
development of a tool for the pygmy blue-tongue skink, Tiliqua adelaidensis,
using animal-based measures
General comment
This article addresses an important topic, and is generally well-written and well set out. Several of its messages are important and echo current themes in reptile welfare. There is much to commend about this paper, and with some refinement I believe it would make a good article for Animals. One of these important messages is that ‘resource-based’ welfare assessments are inadequate (i.e. presumption that welfare is good because a cage has ‘good’ heating, lighting, and furnishings etc). Few keepers realise this and the authors make that point well.
In places the article is too presumptuousness rather than providing fact-based statements. For example, the authors occasionally seem to imply that the current status with reptile welfare assessment is, essentially, ‘not good’ or needs ‘validation’ (which derogates current approaches) and that they have the answers. Of course, considerably more can and probably will be added to the suite of assessments that we already use, but it is essentially the current behavioural assessments that the authors use for their own method. Also, the authors over-claim the role of zoos in education and conservation.
I should suggest that the authors also obtain the following volume: Health and Welfare of Captive Reptiles, Chapman & Hall, publishers, for greater background and context on reptile welfare and its assessment.
The tool that the authors propose is acceptable, and borrows heavily from existing work on reptile welfare assessment. I think the new tool does contribute another approach that will help evaluate reptile welfare, but it does not and should not be ‘sold’ as replacing what we have.
The authors test their method on one species and do mention that it is a pilot study. However, they also propose the method for all reptiles and further suggest that their method is transposable to other classes too. Because of the highly limited scope of this paper, I think it would be better for the authors to increase emphasis that their blue tongued skink example is a pilot assessment.
Specific comments
Lines 12 – 14
This sentence is too sweeping. Whilst it is correct that signs of pain or disease and welfare assessments general can be difficult for many people to assess, there are a growing number of experts who are quite capable of making such assessments.
Lines 18 – 19
Not clear, needs rewording.
Lines 18 – 19
Reads: “This tool will aid in conservation efforts to establish and improve welfare standards for the pygmy blue-tongue skink, and can serve as a basis for future tool design for use in other reptile species.”
Sweeping statement, this time implying that conservation efforts for this one species are successful. If the authors intend to make conservation issues a positive outcome measure for their work, then where is the independently verified evidence that captive-breeding has ‘conserved’ this species?
Line 27
Explain ‘resource-based’ and ‘animal-based’ at this stage for unfamiliar readers.
Lines 30 – 32
See ‘General comments’ above.
Lines 40 – 44
The authors need to be cautious about over-claiming the conservation and education roles of zoos. Apart from a substantial lack of evidence to support their view, criticism from within the zoo community confirms that over-claiming is a recognised problem (e.g. Moss, A. 2018, Is education working? Conservation Education).
Line 50
Needs rewording slightly. Reptiles are essentially ectothermic, and a few are more endothermic than others. All animals, including ‘strict’ endotherms are somewhat ectothermic too, being dependent on environmental temperatures to some degree.
Lines 51 – 52
I’m not sure the claim on popularity in the UK stands – the USA may have a greater proportion of reptiles per human population.
Line 52
Reptiles do not ‘make up 21% of the live animal trade’. The most reliable data indicate 40 – 50 million pet fish and only 900,000 – 1.1 m reptiles in the UK (PFMA, 2017). The Robinson et al reference cited is not reliable on several fronts.
Lines 53 – 55
Mortality rates in reptiles are partly due to the reasons cited by the authors and partly due to inherent low adaptability to artificial environments – many references conclude this point.
Lines 55 – 56
‘Stoic’ is a poor and outdated term in relation to reptiles.
Lines 58 – 59
This is over-claiming - an assessment tool will ‘prevent’ suffering? That is an oxymoron – because as soon as an observer identified poor welfare the ‘prevention’ component has failed.
Lines 71 – 72
Snakes and lizards are currently most popular? How known? In the UK the numbers are equal between snakes, lizards and turtles (PFMA, 2016-17).
Lines 85 – 86
Reads: “Accordingly, less than 1% of the social behaviors utilized by reptile species is currently known.”
How do the authors quantify what we don’t know?
Lines 89 – 90
Says who?
Lines 90 – 91
Again, says who? There’s a lot out there already of you know where to look.
Lines 91 – 92
There have been a lot of reptile oriented welfare papers in more recent years.
Lines 99 – 101
Some good messages in this section, but who decides what is ‘optimum’?
Lines 102 – 105
Good messaging.
Lines 159 – 160
Probably true, but some behaviour, e.g. sleep, may not involve any feeling at certain times.
Lines 157 – 364
This is a good subject review section.
Lines 374 – 377
Captivity-stress-related abnormal maladaptive stereotypies (i.e. of the kind observed in some mammals, pacing, head swaying etc) are not known in reptiles.
Lines 385 – 386
Basking can also occur for a variety of negative reasons including disease (behavioural fever), stress (emotional fever), too low ambient temperatures, too small basking sites, under-representation of other provisions leading to ‘hyperbasking’, and other reasons.
Lines 402 – 403
Acute stress means short-term stress, and reptiles, due to their slow metabolic rate and delayed disease onset times, do not lose physical conditions quickly. Loss of physical condition (e.g. emaciation) implies chronic stress.
Lines 403 – 404
Obesity in some cases yes, otherwise see comment for Lines 402 – 403 above.
Lines 407 – 408
Body score helpful as one limited measure, it is not useful for holistic welfare assessment.
Lines 427 – 428
I do not believe that any zoo meets ‘all’ welfare standards, and certainly the author of the cited ref (67, Draper), whilst a genuine welfare proponent, is not expert in the relevant field of reptile welfare assessment.
Line 437
Reads: ‘adaptions’
Do authors mean ‘adaptations’?
Lines 471 - 473
Reads: “Despite this, due to their knowledge and time spent with the animals, keepers are the best candidates for recognizing abnormal behavior, developing enrichment strategies and assessing the welfare of animals under their care [19].”
This is highly questionable. In an ideal world, maybe this would be true but in the real world it is not. In my long experience, keepers are generally very poorly informed (whether zoo or private) and resistant to objective information. Much practice is what we call ‘folklore husbandry’.
Lines 485 – 493
Mention also that most snakes and many other reptiles are crepuscular or nocturnal, which can compromise comprehensive assessment if assessments are performed diurnally. Obviously, it is better to assess nocturnal animals at night.
Lines 540 – 541
See other comments re over claiming and conservation.
Lines 541 – 543
Reads: “even less expressive species of reptile, such as a species of snake”
What does this mean?
Table 1. Reads: “It is recommended that an enclosure size of 0.2m2per 0.1m of total length is sufficient for terrestrial lizards. The pygmy skink is 90mm snout-to-vent in length so requires a minimum of 0.2m2enclosure space.”
This seems very a small enclosure recommendation. The more recent algorithm of 10 x body size diameter, with an absolute minimum of 2500px primary linear dimension is the better absolute minimum (https://www.frontiersin.org/articles/10.3389/fvets.2018.00151/full).
Author Response
Comment 1
Lines 12 – 14: This sentence is too sweeping. Whilst it is correct that signs of pain or disease and welfare assessments general can be difficult for many people to assess, there are a growing number of experts who are quite capable of making such assessments.
Response 1
We agree with the reviewer’s assertion that there are many who can do this with considerable skill. We have modified this sentence to not suggest that no-one is able to do this well but that there are reasons why it may be challenging. This now reads, L 12-13: ‘Husbandry of reptiles is complex; signs of pain or disease can be challenging to recognize, and behavior is not always well understood.’
Comment 2
Lines 18 – 19: Not clear, needs rewording.
Reads: “This tool will aid in conservation efforts to establish and improve welfare standards for the pygmy blue-tongue skink, and can serve as a basis for future tool design for use in other reptile species.”
Sweeping statement, this time implying that conservation efforts for this one species are successful. If the authors intend to make conservation issues a positive outcome measure for their work, then where is the independently verified evidence that captive-breeding has ‘conserved’ this species?
Response 2
On reflection, we accept that this statement was an overstatement. The focus of this work is on welfare assessment so we have reworded to remove a conservation angle.
Comment 3
Line 27: Explain ‘resource-based’ and ‘animal-based’ at this stage for unfamiliar readers.
Response 3
We agree that this probably needs greater definition but have expanded on this in the main paper at lines 111-118, given the limited word count available in the abstract. These lines in Section 3 now read ‘Resource-based measures are recorded in the animals’ environment. They are quantitative, highly repeatable across different observers, and easy to record. However, these measures may not be correlated with the actual affective state or condition of an animal. For example, a lizard may have optimum lighting and space in its enclosure but may be suffering from a debilitating genetic disease. Conversely, animal-based indicators are measured directly in animals, using a combination of physiological, behavioral and health variables
Comment 4
Lines 30 – 32: See ‘General comments’ above.
Response 4
As described above, greater definition of these terms have been provided in section 3 of the manuscript.
Comment 5
Lines 40 – 44: The authors need to be cautious about over-claiming the conservation and education roles of zoos. Apart from a substantial lack of evidence to support their view, criticism from within the zoo community confirms that over-claiming is a recognised problem (e.g. Moss, A. 2018, Is education working? Conservation Education).
Response 5
We agree that the conservation role of zoos may be overclaimed. We do not believe that the statement as written indicates that zoos are successful in this mission, but we do believe that it is in the remit of most zoos to have a conservation role, and that they do indeed promote this when marketing. This has been rewritten to reflect our meaning that this is generally a mission of zoos (be it successful or not….). This now reads. L 39-40: ‘Zoos are increasingly positioning themselves and acting as conservation and education institutions.’
Comment 6
Line 50: Needs rewording slightly. Reptiles are essentially ectothermic, and a few are more endothermic than others. All animals, including ‘strict’ endotherms are somewhat ectothermic too, being dependent on environmental temperatures to some degree.
Response 6
This sentence has been reworded to include the phrase ‘primarily ectothermic’ to allow for the possibility that thermogenesis may be used by some species, L 65.
Comment 7
Lines 51 – 52: I’m not sure the claim on popularity in the UK stands – the USA may have a greater proportion of reptiles per human population.
Response 7
We thank the reviewer for this comment and agree that ‘popular’ is an incorrect descriptor. We have sourced some recent statistics from the UK and US on pet reptile ownership. This has been added at lines 44-45.
Comment 8
Line 52: Reptiles do not ‘make up 21% of the live animal trade’. The most reliable data indicate 40 – 50 million pet fish and only 900,000 – 1.1 m reptiles in the UK (PFMA, 2017). The Robinson et al reference cited is not reliable on several fronts.
Response 8
Reference to the live animal trade data has now been removed, as a result of the preceding comment. Consequently the Robinson et al reference is now longer cited here.
Comment 9
Lines 53 – 55: Mortality rates in reptiles are partly due to the reasons cited by the authors and partly due to inherent low adaptability to artificial environments – many references conclude this point.
Response 9
Of course. We have added some information to this effect at lines 47.
Comment 10
Lines 55 – 56:‘Stoic’ is a poor and outdated term in relation to reptiles.
Response 10
The word ‘stoic’ has been removed from this sentence.
Comment 11
Lines 58 – 59: This is over-claiming - an assessment tool will ‘prevent’ suffering? That is an oxymoron – because as soon as an observer identified poor welfare the ‘prevention’ component has failed.
Response 11
We agree that this wording is clumsy and have reworded to ‘Reliable tools to objectively assess affective state, and consequently welfare, of reptiles would better allow those caring for reptiles to identify deficiencies in wellbeing and to evaluate management strategies employed to improve health and welfare, L 52-55.
Comment 12
Lines 71 – 72: Snakes and lizards are currently most popular? How known? In the UK the numbers are equal between snakes, lizards and turtles (PFMA, 2016-17).
Response 12
This is true and the phrase has been reworded to ‘These are currently the most common species kept as pets’, L 69.
Comment 13
Lines 85 – 86
Reads: “Accordingly, less than 1% of the social behaviors utilized by reptile species is currently known.”
How do the authors quantify what we don’t know?
Response 13
We apologise for this clumsy wording. The statement should read ‘Accordingly, social systems are currently known in less than 1% of lizard species.’ This has now been amended, L 82.
Comment 14
Lines 89 – 90: Says who?
Response 14
This is our impression from performing a literature review and is in comparison to the body of literature on physiological measures of stress in other vertebrates. We have attempted to clarify this statement at lines 87-94.
Comment 15
Lines 90 – 91: Again, says who? There’s a lot out there already of you know where to look.
Response 15
As above, this statement was made based on our impressions when searching the literature and from chatting to those who work with reptiles. We agree that there is a wealth of information available and there certainly seems to be increasing focus on reptiles welfare but reptiles still appear understudied in comparison with mammals. This has been reworded at lines 87-94.
Comment 16
Lines 91 – 92: There have been a lot of reptile oriented welfare papers in more recent years.
Response 16
We agree and have added some lines into the manuscript to reflect that research interest in this area does appear to have increased. See lines 93-94, ‘However, a database search does suggest that research efforts into reptile welfare have increased in recent years and it is hoped that this trend continues.’
Comment 17
Lines 99 – 101: Some good messages in this section, but who decides what ‘optimum’ is?
Response 17
The point here was a generic example to highlight that the provision of resource-based measures to a prescribed or ‘best-practice’ standard, in accordance with the range of guidelines available, does not necessarily mean that the animal is in a state of ‘good’ welfare. We do not believe this requires further elaboration as optimum in this case is a hypothetical goal.
Comment 18
Lines 102 – 105: Good messaging.
Response 18
Thanks.
Comment 19
Lines 159 – 160: probably true, but some behaviour, e.g. sleep, may not involve any feeling at certain times.
Response 19
We agree with the reviewer’s comments of course, but feel that the statement as it stands is correct as it suggests that behaviour alone is not enough to gauge affective state, but can be used as a guide. This would appear to cover behaviours such as sleep, where other measures would need to be considered (such as physiology or measurement of a change in sleep duration), to gauge affective state. The statement reads ‘Behavior alone is not directly indicative of affective state but provides insight into how an animal feels’, L 185-186.
Comment 20
Lines 374 – 377: Captivity-stress-related abnormal maladaptive stereotypies (i.e. of the kind observed in some mammals, pacing, head swaying etc) are not known in reptiles.
Response 20
Thank you for alerting us to this. We have clarified that maladaptive stereotypies have not been observed in reptiles at lines 445-446.
Comment 21
Lines 385 – 386: Basking can also occur for a variety of negative reasons including disease (behavioural fever), stress (emotional fever), too low ambient temperatures, too small basking sites, under-representation of other provisions leading to ‘hyperbasking’, and other reasons.
Response 21
Thank you for this suggestion. We have added more information on basking at lines 462-468 which now reads ‘Basking may be an indicator of negative welfare states since reptiles use behavior in modulating their response to both internal (physiological), and external conditions. For example, in response to ‘true’ fever or emotional fever caused through exposure to stressful events such as handling, reptiles preferentially seek warmer temperatures. Deficiencies in the environment such as low ambient temperatures, inadequate access to basking sites through competition or overcrowding, or the under-provision of other ‘cage facilities’ leading to ‘hyperbasking’ can all increase the time spent basking.
Comment 22
Lines 402 – 403: Acute stress means short-term stress, and reptiles, due to their slow metabolic rate and delayed disease onset times, do not lose physical conditions quickly. Loss of physical condition (e.g. emaciation) implies chronic stress.
Response 22
Lines 484-486 have been reflected to make this clearer. This now reads: ‘Chronic and progressive stress which leads to a prolonged increase in corticosterone results in muscle degeneration and growth suppression. This consequently predisposes to emaciation. However, some reptiles that are undergoing chronic stress will present with obesity’
Comment 23
Lines 403 – 404: Obesity in some cases yes, otherwise see comment for Lines 402 – 403 above.
Response 23
We agree and have reworded as above.
Comment 24
Lines 407 – 408: Body score helpful as one limited measure, it is not useful for holistic welfare assessment.
Response 24
We agree with the reviewer that body condition on its own is not appropriate for welfare assessment but believe that the current wording does not suggest we are proposing this. We are merely proposing its inclusion in a welfare assessment tool and hopefully providing justification for our inclusion of it later in the paper.
Comment 25
Lines 427 – 428: I do not believe that any zoo meets ‘all’ welfare standards, and certainly the author of the cited ref (67, Draper), whilst a genuine welfare proponent, is not expert in the relevant field of reptile welfare assessment.
Response 25
We agree with the reviewer’s point but believe that the paper referenced was only considering legislated standards not optimal conditions that might be required for reptiles. We have hence revised these lines to make this meaning clearer. Changes have been made at lines 475-479 which now reads, ‘Draper & Harris 2012 reported that only 24% of 192 British zoos inspected met all legislated welfare standards [79]. Similar findings in other parts of the world might be expected. There are increasing efforts in many jurisdictions to ensure welfare standards are upheld’. Lines 512-514.
Comment 26
Line 437: Reads: ‘adaptions’. Do authors mean ‘adaptations’?
Response 26
Thank you for highlighting this. Now amended to ‘adaptations’. L 529.
Comment 27
Lines 471 – 473:
Reads: “Despite this, due to their knowledge and time spent with the animals, keepers are the best candidates for recognizing abnormal behavior, developing enrichment strategies and assessing the welfare of animals under their care [19].”
This is highly questionable. In an ideal world, maybe this would be true but in the real world it is not. In my long experience, keepers are generally very poorly informed (whether zoo or private) and resistant to objective information. Much practice is what we call ‘folklore husbandry’.
Response 27
We agree and have changed the language slightly to reflect that we believe they should be equipped to be able to do this, rather than may actually be the ‘best’ people at doing this. These lines now read ‘Despite this, due to their knowledge, experience, and time spent with the animals they were evaluating, keepers should be best placed to recognize abnormal behavior, develop enrichment strategies, and assess the welfare of animals under their care [25]. Consequently, they must be equipped with the tools to formally perform these activities successfully.’
Comment 28
Lines 485 – 493: Mention also that most snakes and many other reptiles are crepuscular or nocturnal, which can compromise comprehensive assessment if assessments are performed diurnally. Obviously, it is better to assess nocturnal animals at night.
Response 28
Excellent point. Mention of this has been made at lines 594-595.
Comment 29
Lines 540 – 541: See other comments re over claiming and conservation.
Response 29
This section has been amended in light of another reviewer’s comments. We believe we have softened the statements in relation to conservation in general.
Comment 30
Lines 541 – 543: Reads: “even less expressive species of reptile, such as a species of snake”.
What does this mean?
Response 30
This should read ‘behaviorally expressive’ and has been amended.
Comment 31
Table 1. Reads: “It is recommended that an enclosure size of 0.2m2per 0.1m of total length is sufficient for terrestrial lizards. The pygmy skink is 90mm snout-to-vent in length so requires a minimum of 0.2m2enclosure space.”
This seems very a small enclosure recommendation. The more recent algorithm of 10 x body size diameter, with an absolute minimum of 2500px primary linear dimension is the better absolute minimum (https://www.frontiersin.org/articles/10.3389/fvets.2018.00151/full).
Response 31
Thank you for alerting us to this paper. We have now referred to it in the table. The requirements seem fairly similar e.g 1 m x 0.2 m to make an area of 0.2m2 versus a minimum of 1m.
Round 2
Reviewer 1 Report
Thank you for answering so well all my comments and also the ones from the other reviewer.